# Human resources for nephrology in South Africa: A mixed-methods study

**Muhammed Hassen**[1], **Elize Archer**[2], **Adriano Pellizzon**[1], **Usuf M. E. Chikte**[3], **Mogamat Razeen Davids**[1]*

1 Division of Nephrology, Stellenbosch University and Tygerberg Hospital, Cape Town, South Africa,
2 Centre for Health Professions Education, Stellenbosch University, Stellenbosch, South Africa, 3 Division of Health Systems and Public Health, Department of Global Health, Stellenbosch University, Stellenbosch, South Africa

* mrd@sun.ac.za

## Abstract

### Introduction

The global nephrology workforce is shrinking and, in many countries, is unable to meet healthcare needs. Accurate data pertaining to human resources in nephrology in South Africa is lacking. This data is critical for the planning and delivery of renal services and the training of nephrologists in South Africa to meet the challenge of the growing burden of chronic kidney disease.

### Methods

A cross-sectional study of adult and paediatric nephrologists currently delivering nephrology services in South Africa was conducted. Participants were identified using various data sources, including the register of the Health Professions Council of South Africa. This cohort of doctors was described in terms of their demographics and distribution. A survey was then conducted among these nephrologists to collect additional information on their training, scope of practice, job satisfaction, challenges and future plans. Finally, two focus group interviews were conducted to probe themes identified from the survey data.

### Results

A total of 120 adult nephrologists and 22 paediatric nephrologists were identified (an overall density of 2.5 per million population). There is a male predominance (66%) and the median age is 45 years. The bulk of the workforce (128 nephrologists, 92%) is distributed in three of the nine South African provinces, and two provinces have no nephrologist at all. The survey was completed by 57% of the nephrologists. Most reported positive attitudes to their chosen profession; however, 35 nephrologists (43%) reported an excessive workload, 9 (11%) were planning emigration and 15 (19%) were planning early retirement. A higher frequency of dissatisfaction regarding remuneration (39% vs. 15%) and unsatisfactory work conditions (35% vs. 13%) was observed amongst nephrologists working in the public sector compared to the private sector. A total of 13 nephrologists participated in the focus group interviews. The themes which were identified included that of a rewarding profession, an overall

**Data Availability Statement:** All relevant data are within the paper and its Supporting Information files.

**Funding:** The authors received no specific funding for this work.

**Competing interests:** The authors have declared
that no competing interests exist.

shortage of nephrologists, poor career planning, a need for changes to nephrologists' training, excessive workloads with inadequate remuneration, and challenging work environments.

## Conclusion

There are insufficient numbers of nephrologists in South Africa, with a markedly uneven distribution amongst the provinces and healthcare sectors. Qualitative data indicate that South African nephrologists are faced with the challenges of a high workload, obstructive policies and unsatisfactory remuneration. In the public sector, a chronic lack of nephrologist posts and other resources are additional challenges. A substantial proportion of the workforce is contemplating emigration.

## Introduction

Chronic kidney disease (CKD) is an important non-communicable disease that contributes substantially to morbidity and mortality [1]. The burden of CKD is escalating, and recent estimates suggest that between 8% and 16% of the global population is affected, with non-communicable disease being responsible for the majority of CKD cases in developed countries [2]. Developing nations like South Africa have the additional drivers of infectious diseases, injuries, pregnancy-related complications and exposure to environmental toxins that further contribute to the increasing prevalence of CKD [3]. Two recent estimates have reported CKD population prevalences in South Africa of 6.4% and 17.3%, respectively [4, 5].

Resources for medical care are limited in most parts of Africa, resulting in a lack of access to treatment for patients with advanced CKD and end-stage renal disease (ESRD). In South Africa, the prevalence of patients treated with renal replacement therapy (RRT) in 2017 was only 190 per million population (pmp) [6]. This prevalence reflects the inadequate provision of services and access to treatment rather than the burden of disease. Thailand (1515 pmp), Turkey (933 pmp), Brazil (865 pmp), Colombia (672 pmp) and Indonesia (323 pmp) are countries with similar or lesser gross domestic product per capita than South Africa but with much a higher RRT prevalence, highlighting the lack of access to this treatment in South Africa [7].

South Africa is a ranked by the World Bank as an upper-middle-income country [8] with high levels of inequality [9]. A two-tier model of healthcare exists: a well-resourced private sector that mainly serves the 16% of the population who are able to afford medical insurance; and a poorly resourced public sector serving mainly the 84% of the population who are uninsured [10]. The prevalence of RRT is 66 pmp and 855 pmp in the public and private healthcare sectors, respectively [6]. This discrepancy exists due to the rationing of government supported RRT in the public sector whilst greater access is available for members of medical insurance schemes in the private sector [11]. Apart from the disparity in access to RRT that exists between public and private healthcare sectors, there are also striking disparities between the different provinces of South Africa, with two provinces (Limpopo and Mpumalanga) having no public sector dialysis centres at all [12].

Regarding human resources for the delivery of nephrology services, it is of concern that the global nephrology workforce is shrinking despite the growing burden of CKD [13]. Data from the UK, Australia and the USA indicate that contributing factors include decreased interest in a nephrology career amongst medical students and trainees, perceptions of poor remuneration, physician burn-out, increased reliance on immigrant nephrologists and early retirement

[13–15]. In African countries, the nephrology workforce is further jeopardised by emigration to developed countries [16].

The 2019 Global Kidney Health Atlas (GKHA) [17] provides the most current data on the global nephrology workforce and reports that the median density of nephrologists worldwide is 9.95 pmp. Higher densities are found in Europe (24.4–25.6 pmp), North and East Asia (19.5 pmp) and North America (18.1 pmp), and lower densities in Latin America (9.8 pmp), the Middle East (8.1 pmp), Oceania and South-East Asia (5.7 pmp), South Asia (1.2 pmp) and Africa (0.6 pmp). Nephrologist density is associated with country income. Low-income countries report the lowest median numbers (0.2 pmp), followed by lower-middle (1.6 pmp), upper-middle (10.8 pmp) and high (23.2 pmp) income countries.

Stark differences exist across the African continent, with far greater densities in North African than sub-Saharan countries. In a survey on the nephrology workforce which was part of the GHKA project, Osman et al. [18] reported that 9 of the 10 countries worldwide with the lowest nephrologist density are from sub-Saharan Africa. Malawi (0.06 pmp) and Mozambique (0.08 pmp) had the lowest, and Egypt (21.7 pmp) and Tunisia (16.3 pmp) had the highest densities among the African countries.

The African data on human resources for nephrology are mostly based on estimates by senior nephrologists in various countries, rather than on formal publications reporting robust data [18, 19]. In South Africa, there is a similar lack of accurate data. Various estimates of the density of nephrologists range between 1.1 pmp [19] and 1.86 pmp [18]. Detailed and accurate data is critical in planning for the training, employment and retention of nephrologists and for the delivery of effective renal services in the country [20, 21]. This includes information on overall numbers and information on any within-country disparities in nephrologist density related to factors such as geography or healthcare sector (public/private). We, therefore, conducted a study to determine the numbers, demographics and distribution of the nephrologist workforce in South Africa and to probe additional factors impacting on the delivery of nephrology services by this cohort of specialists. These factors included their training, skill sets and the nature of their practices, as well as their challenges, retirement plans and emigration considerations.

## Methods

The research team: MH was a nephrology fellow at the time of the study, AP a recently qualified nephrologist and MRD a nephrologist and the head of an academic division of nephrology. EA is a nurse by training, has a PhD in medical education, and is experienced in qualitative research. UMEC is a dentist by training, and currently the head of an academic department of Global Health. The South African nephrology community is a small one, with most members known to one other; this facilitated access to potential participants. None of the research team was involved in teaching or supervising any participant or had authority over any participant at the time of the study. Most of the members of the research team were based at public sector institutions, except for AP who was based in private practice. MRD also had a very limited involvement in a private sector dialysis centre. Our assumptions, based on prior knowledge and our own experiences in a chronically under-resourced public sector, were that we were likely to find a low overall density of nephrologists and that there was likely to be important within-country disparities as well. Reflexivity was maintained throughout the project by discussing and challenging the assumptions and potential biases of the nephrology members of the team.

A cross-sectional, mixed-methods study of nephrologists in South Africa was undertaken during 2018. There were three parts to the study. First, a number of data sources were used to identify all the active nephrologists working in South Africa and these included the databases

of the Health Professionals Council of South Africa (HPCSA), the Colleges of Medicine of South Africa, the South African Renal Society and the South African Renal Registry. "Active" was defined as being involved in nephrology as a clinician, researcher, administrator or teacher, and "nephrologists" were defined as doctors who were registered with the national regulatory body, the HPCSA, as adult or paediatric nephrologists, or who were about to regis- ter as such, having completed their formal training in nephrology and having passed the exit examination (the Certificate in Nephrology of the College of Physicians or the College of Pae- diatricians). Nephrologists working outside of South Africa were excluded (nine nephrolo- gists), as were those known to have retired (two nephrologists).

The following demographic information was extracted from our data sources: age, sex, eth- nicity, province and healthcare sector (public/private). This was reconciled with information about practicing nephrologists from personal knowledge and personal contacts. Mid-year pop- ulation estimates for 2018 from Statistics South Africa [22] were used to calculate nephrologist densities, and RRT prevalence data for 2017 from the latest report of the South African Renal Registry [6] were used to calculate the ratios of nephrologists to patients on RRT.

The second part of our study involved conducting a survey amongst this cohort of nephrol- ogists to firstly confirm the demographic data obtained from the sources described above, and then to collect additional data on their training, qualifications, nature of practice, skill sets and attitudes towards their practice, as well as their challenges, retirement plans and emigration considerations. Respondents were also asked to indicate their willingness to participate in a focus group discussion (part 3) at a later stage. The survey was developed and managed by MH and MRD using REDCap, a secure web-based application designed to support data capture for research [23]. The distribution list for the survey was compiled from the sources listed above and included all active nephrologists. Before general use, the questionnaire (S1 Appendix) was refined via a pilot survey with 10 local nephrologists. Questionnaires were sent out at regular intervals until the end of the study period.

The third part of the study involved focus group interviews which were conducted at the 2018 South African Renal Congress, held in Johannesburg in October 2018. This biennial meeting is the official congress of the South African Renal Society and is attended by most nephrologists, providing us with an opportunity to gain additional insight into participants' perspectives of the themes identified through the REDCap survey (part 2). We decided that focus group discussions would be an efficient method of data collection, enabling group mem- bers to feed off each other's ideas and permitting flexibility in the discussion while our moder- ators ensured that the overall focus was maintained. Two interviews were conducted at the conference centre, with public sector nephrologists in one session and private sector nephrolo- gists in the other. The participants were selected from the survey respondents who had indi- cated a willingness to participate. Purposive sampling was used to ensure variation in age, sex, province and category of practice (adult or paediatric nephrology). The interviews were mod- erated by MH and MRD and each lasted approximately 60 minutes. The interviews were con- ducted in English, using open-ended questions and prompts (S2 Appendix) to facilitate the discussion and avoided the use of leading questions. Interviews included discussions of the enjoyable aspects of a nephrology career, the adequacy of nephrology training programmes, current challenges and future concerns. Discussion on a particular item was allowed to con- tinue until data saturation occurred. The proceedings were electronically recorded and tran- scribed verbatim. MH and MRD also collected field notes.

Analysis of the quantitative data was performed in Microsoft Excel, with values summarised using counts and percentages, and medians with interquartile ranges (IQR).

Interviews were anonymised and each participant was given a code number. The qualitative data was processed using framework analysis as described by Krueger and Casey [24]. MH and

EA coded the transcripts and identified emerging themes. These themes were further refined by discussion amongst all the authors until consensus was reached. All the important responses were indexed, and selected quotations are reported verbatim in the results. There was one instance where we were concerned that a participant might be identifiable through the example provided. The matter was discussed with the participant concerned, who confirmed that the contribution could be used.

Ethical approval for the project was obtained from the Human Research Ethics Committee of Stellenbosch University (reference number S16/05/094). All participants provided informed consent. In the case of the survey, it was clearly communicated at the beginning of the survey that completion thereof indicated consent to participate (see S1 Appendix), and in the case of the focus group interviews, verbal consent was provided. All participants were aware that they were part of a research project which would lead to a published report.

## Results

### Quantitative data

The South African population in 2018 was estimated at 57.7 million [22]. A total of 120 adult and 22 paediatric nephrologists were identified, yielding an overall density of 2.5 nephrologists pmp. In 2017, there were 10 744 patients treated with RRT in South Africa, an average of 76 patients per nephrologist [6]. Demographic details of the cohort are shown in Table 1. The median age is 45 years (IQR 40–54 years) and there is a male predominance (66%). Female nephrologists (median age 41.5 years, IQR 38–47.5 years) were younger than their male counterparts (median age 48.0 years, IQR (42–58 years) and had more recently qualified as

**Table 1. Demographic details of nephrologists in South Africa.** Abbreviations: IQR, interquartile range.

| | Male, n = 94 | Female, n = 48 | Total, n = 142 |
|---|---|---|---|
| **Age (years), median (IQR)** | 48 (42–58) | 41.5 (38–47.5) | 45 (40–54) |
| **Ethnicity, n (%) *** | | | |
| Black African | 16 (17%) | 8 (16%) | 24 (17%) |
| White | 36 (38%) | 19 (40%) | 55 (39%) |
| Mixed ancestry ("Coloured") | 8 (9%) | 1 (2%) | 9 (6%) |
| Indian/Asian | 34 (36%) | 19 (40%) | 53 (37%) |
| Not specified | 0 (0%) | 1 (2%) | 1 (1%) |
| **Speciality, n (%)** | | | |
| Adult nephrologist | 85 (90%) | 35 (73%) | 120 (85%) |
| Paediatric nephrologist | 9 (10%) | 13 (27%) | 22 (15%) |
| **Province, n (%)** | | | |
| Gauteng | 37 (39%) | 24 (50%) | 61 (43%) |
| Western Cape | 24 (26%) | 13 (27%) | 37 (26%) |
| KwaZulu-Natal | 23 (25%) | 7 (15%) | 30 (21%) |
| Free State | 3 (3%) | 3 (6%) | 6 (4%) |
| Eastern Cape | 6 (6%) | 0 (0%) | 6 (4%) |
| Northern Cape | 1 (1%) | 0 (0%) | 1 (1%) |
| Limpopo | 0 (0%) | 1 (2%) | 1 (1%) |

* Ethnic groups are based on the erstwhile South African Population Registration Act (Act No.30 of 1950). Currently, Statistics South Africa, the national statistical service of South Africa, asks citizens to self-classify into one of five categories: Black African, Coloured, White, Indian/Asian and Other/Unspecified. Coloured refers to South Africans of mixed ancestry.

nephrologists (median of 5 vs. 11 years). In terms of ethnicity, most of the nephrologists were White or Indian/Asian, with Black nephrologists comprising only 16% of the workforce. Nephrologists of African and Indian/Asian descent were more recently qualified (median durations of 5.0 and 7.0 years, respectively) compared to those of mixed ancestry or Whites (8.0 and 9.5 years, respectively). Most nephrologists were South African citizens (91.5%, n = 130), while six were citizens of other African countries (4.2%), four were citizens of European countries (2.8%) and two were citizens of countries in Asia (1.4%).

Most of the adult nephrologists (60%, n = 72) are working in the private sector, while most of the paediatric nephrologists (68%, n = 15) are based in the public sector. Fig 1 illustrates the geographical distribution. There are gross disparities, with 128 nephrologists (92%) working in three of the nine provinces (Gauteng, Western Cape and KwaZulu-Natal), only one each in Limpopo and the Northern Cape, and none in the North West and Mpumalanga.

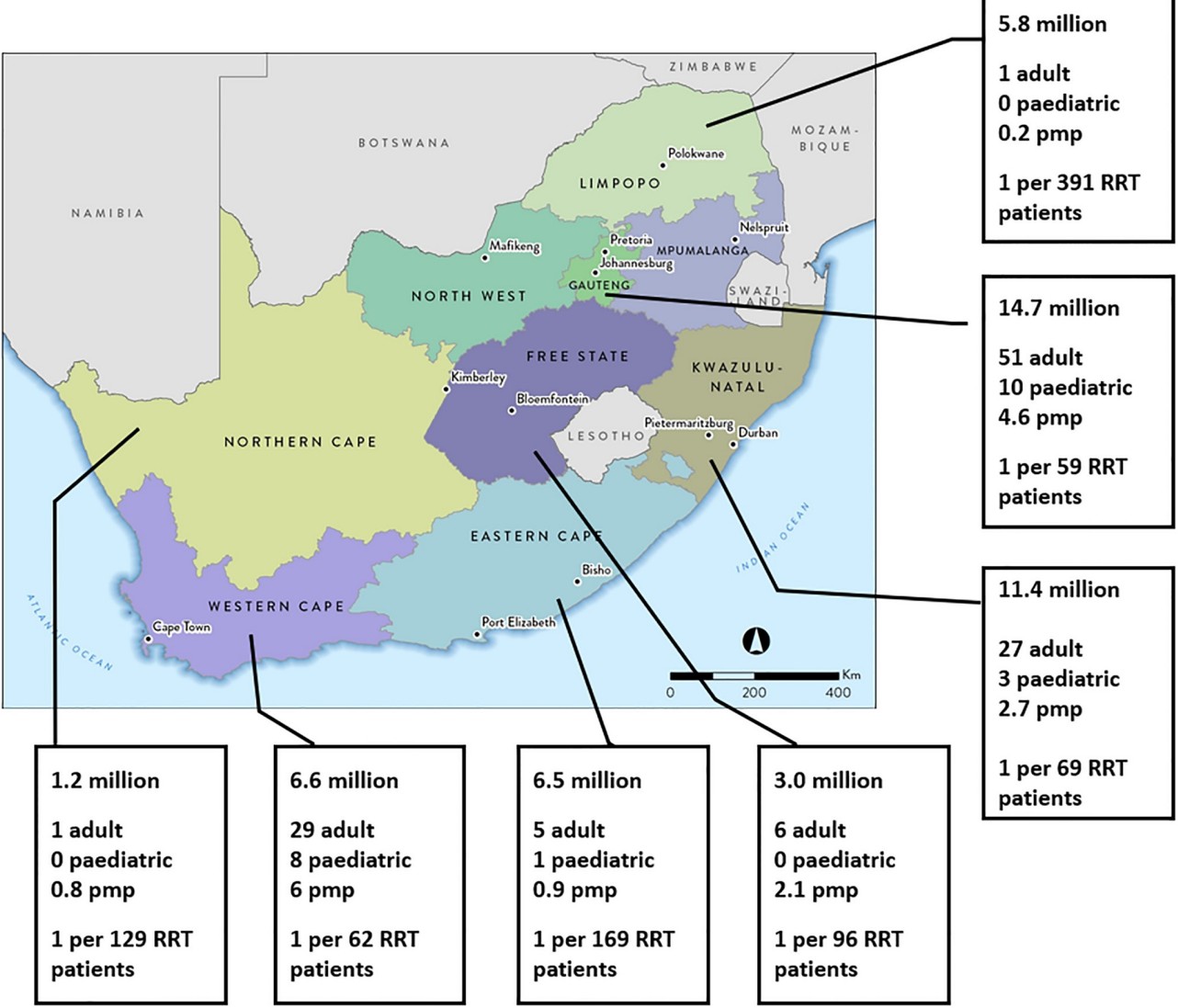

**Fig 1. Distribution and density of nephrologists in South Africa.** The data panel for each province lists, in order, population of the province in millions, number of adult nephrologists, number of paediatric nephrologists, nephrologist density (per million population) and ratio of nephrologists to patients on renal replacement therapy. Abbreviations: pmp, per million population; RRT, renal replacement therapy.

**Table 2. Survey responses of nephrologists working in the public and private healthcare sectors in South Africa.** Abbreviations: IQR, interquartile range.

| | Public (n = 41) | Private (n = 40) | Total (n = 81) |
|---|---|---|---|
| **Employment, n (%)** | | | |
| Full time | 39 (95%) | 37 (92%) | 76 (94%) |
| Part time | 2 (5%) | 3 (8%) | 5 (6%) |
| **Workload, n (%)** | | | |
| Adequate | 18 (44%) | 24 (60%) | 42 (52%) |
| Insufficient | 2 (5%) | 1 (3%) | 3 (4%) |
| Excessive | 21 (51%) | 15 (37%) | 36 (44%) |
| **Remuneration, n (%)** | | | |
| Satisfactory | 23 (56%) | 32 (80%) | 55 (68%) |
| Unsatisfactory | 16 (39%) | 7 (18%) | 23 (28%) |
| **Work environment, n (%)** | | | |
| Satisfactory | 27 (66%) | 34 (85%) | 61 (75%) |
| Unsatisfactory | 14 (34%) | 5 (13%) | 19 (23%) |
| **Considering emigration, n (%)** | 2 (5%) | 7 (18%) | 9 (11%) |
| **Considering early retirement (<65 years), n (%)** | 5 (12%) | 10 (26%) | 15 (19%) |

The survey results are shown in Table 2. The response rate was 57%. The demographics of the 81 survey respondents closely matched that of the full cohort of nephrologists with respect to age and sex (S1 Table). Every province having a nephrologist was represented in the survey and both adult and paediatric nephrologists were among the respondents. Four of the respondents were not South African citizens, with two being from other African countries and two from Eurasia.

Respondents completed their nephrology training at six of the eight medical schools in South Africa, with no nephrologist training at Walter Sisulu University or the University of Limpopo. Nearly 80% of respondents practiced in the province where they received their training. Most trained in the Johannesburg area, at hospitals associated with the University of the Witwatersrand (28 respondents, 35%). There were 10 respondents who reported receiving some nephrology training abroad, with three of those undertaking formal fellowships of 4–24 months, supported by the International Society of Nephrology.

Regarding their skills, most nephrologists felt competent to place temporary haemodialysis catheters (89%) and perform native kidney (89%) and allograft (78%) biopsies. Fewer reported competence with respect to performing renal ultrasound (54%), ultrasound assessment of the vascular access (32%), bedside peritoneal dialysis catheter insertion (53%) and placement of tunnelled haemodialysis catheters (22%). More nephrologists reported undertaking self-initiated training in ultrasound procedures than was received during their formal nephrology training programmes (Fig 2). Similar findings were noted among the adult and paediatric nephrologists (S1a and S1b Fig).

Five (6%) respondents indicated that they worked part time (less than 40 hours per week), with four of the five being female. Reasons given included family commitments, providing more time for research, having a part-time post-retirement contract and the lack of full-time posts in the public sector. There were 19 public sector nephrologists who did some work in the private sector, and six private sector nephrologists who did sessional work in public sector facilities. Most of the working hours were spent on clinical nephrology, with teaching and research performed mainly by public sector nephrologists (Fig 3).

Most respondents (95%) indicated an inclination to recommend nephrology as a career to junior doctors or students. Whilst most were happy with their remuneration, workload and

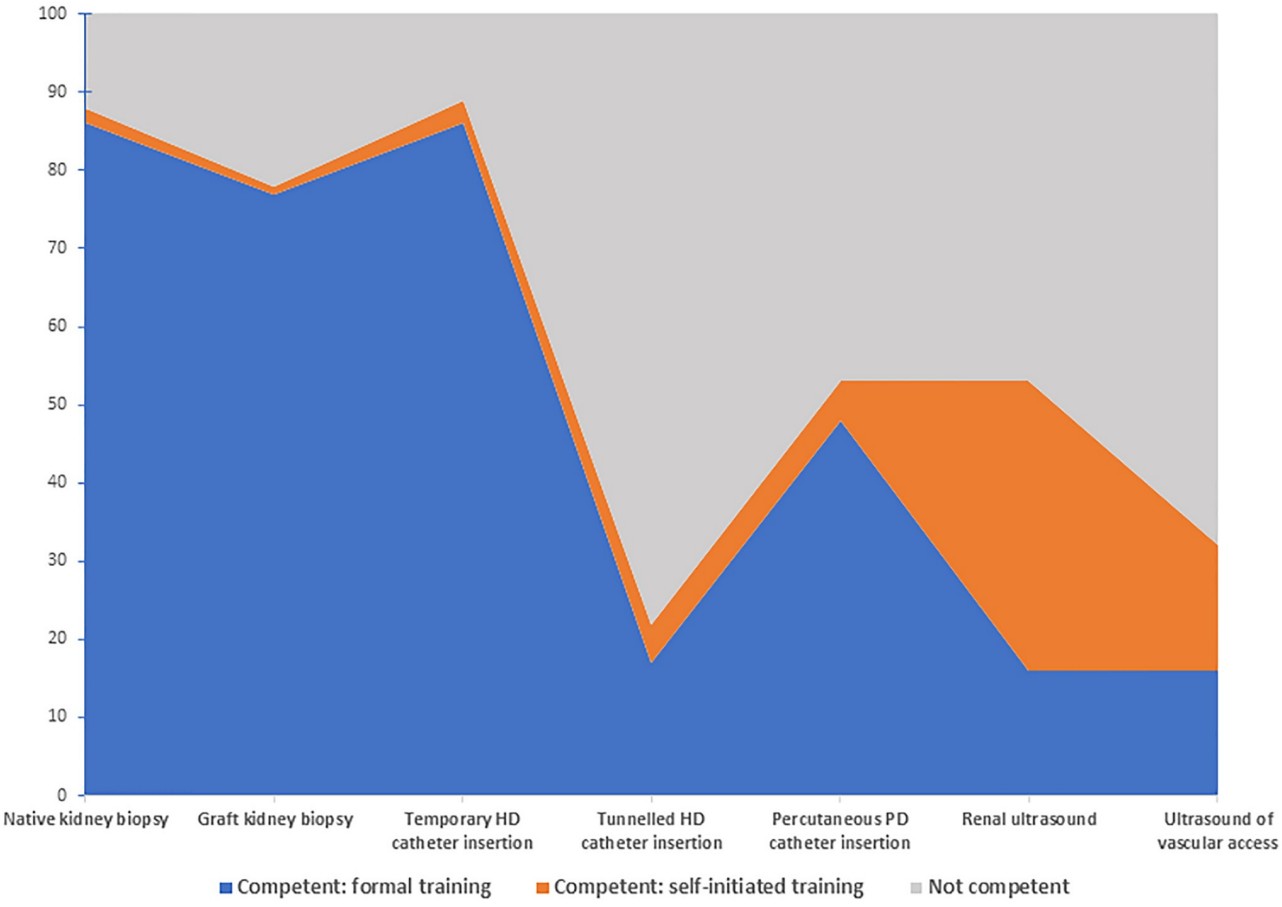

**Fig 2. Percentage of nephrologists reporting competency at various procedures, attained either during formal nephrology training or via self-initiated training.** Abbreviations: HD, haemodialysis; PD, peritoneal dialysis.

working environment, 23 (28%) were unhappy with remuneration, 19 (23%) were unhappy with their working environment and 36 (44%) reported excessive workloads. Unhappiness with remuneration, work environments and excessive workloads were more frequently reported by public sector nephrologists; however, nephrologists working in the private sector were more likely to consider emigration or early retirement. There were nine nephrologists (11%), seven from the private sector and two from the public sector, who planned to emigrate prior to retirement, citing political concerns (four respondents), concerns regarding their children's education (three respondents), better career opportunities abroad (two respondents) and the high crime rate in South Africa (two respondents) as reasons for immigration. Australia was the country most often cited as a possible destination (three respondents) and the median time to planned emigration was 5 years.

There were 15 respondents (19%), 10 from the private sector and 5 from the public sector, who planned to retire early (before the age of 65), with burnout being the most common reason given (six respondents). Being available to support their family (three respondents), seeking more leisure time (two respondents), having attained financial security (two respondents), bureaucratic workplace policies (two respondents) and poor health (one respondent) were the other reasons given. The youngest age of planned retirement was 50 years (two respondents).

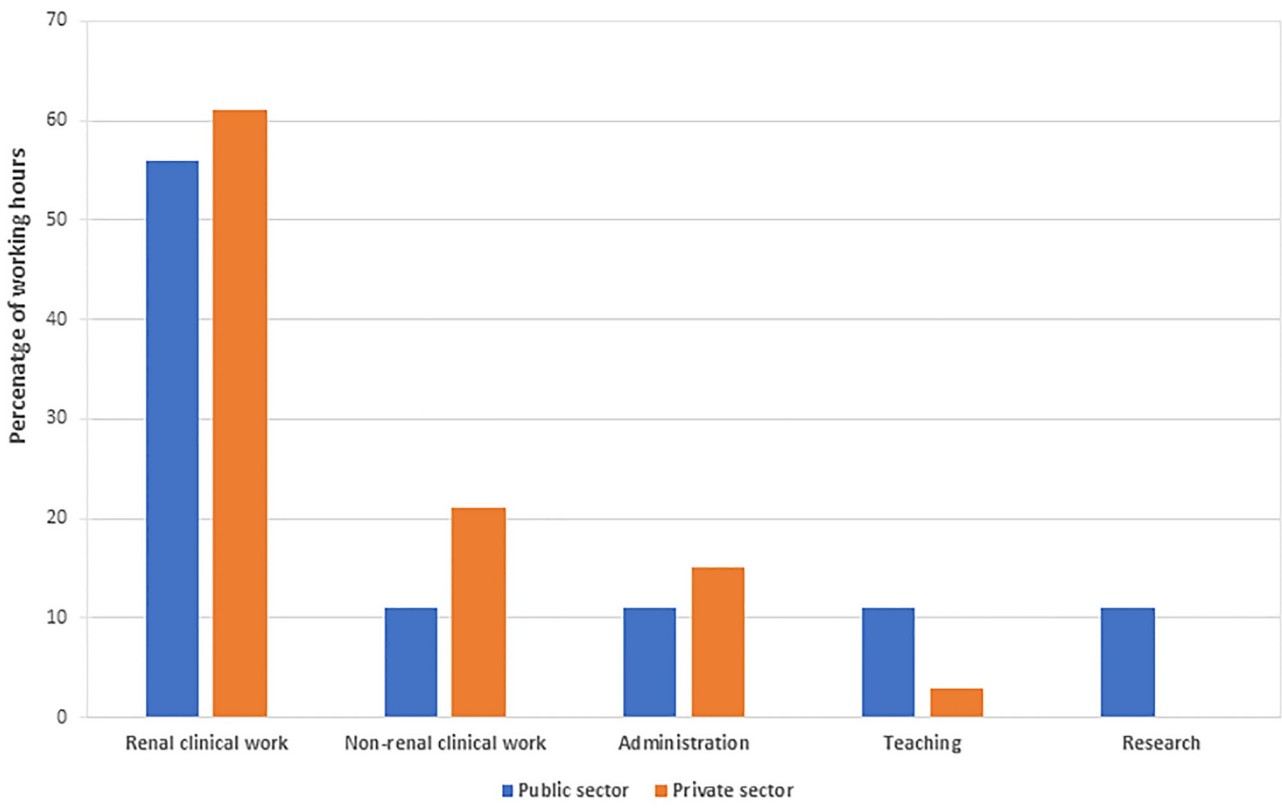

**Fig 3. Percentage of time spent per week in each domain by nephrologists in the public and private sectors.**

## Qualitative data

Thirteen nephrologists participated in the focus group interviews, eight in the public sector interview and five in the private sector interview. The median age of these participants was 44 years (range 36–64) and nine (69%) were male. Participants were from six of the seven provinces in South Africa which had a nephrologist; no nephrologist from the Free State province participated. Three paediatric nephrologists participated, all based in the public sector. The qualitative data from the survey and the focus group interviews are discussed under themes that were identified during the thematic analysis. The following codes apply to the quotes that are reported: AN = adult nephrologist, PN = paediatric nephrologist, F = Female, M = Male.

**Theme 1: A rewarding profession.** Aspects of their practice that nephrologists found satisfying included the clinically and intellectually challenging aspects of nephrology, the diversity of clinical practice, the academic components (with particular mention of teaching), the close and long-term doctor-patient relationships, the bonds among the nephrology community and the ability to heal or "*make a big difference to our patients*". Public sector nephrologists had productive relationships with colleagues and valued the inter-professional collaboration, but there was less consensus on the sense of community in the private sector group, with some commenting on feeling "*isolated*" and remarking on the "*lack of being able to bounce ideas off other people*".

"*. . . there have been some hectic days. When we win and they get transplanted and it's a good outcome, that's really nice.*"

(Public PN, F)

"... *I can be in ICU and then in the outpatient department, and then later the procedure room ... interacting pretty much throughout the whole hospital. So, I will speak to the obstetrician and the surgeon and the psychiatrist in the same day [laughter]*"

(Public AN, M)

"*It's rare that in other specialities where you will see a group of patients a few times a week, almost every day. You know their histories, you know a lot of their social life, you know what's going on, what their hobbies are. For many of our patients, they become our friends.*"

(Private AN, M)

*You put them onto dialysis, you get them through, they get well, you get them transplanted...
and maybe they even get pregnant [laughter]. It's very nice to see that, restoring normal life.*"

(Public AN, M)

"*My passion is for research, and I really enjoy teaching, but it's just [too little] time*"

(Public AN, M)

**Theme 2: A lack of nephrologists and a lack of career planning.** Subthemes of a geographic and sectorial maldistribution, emigration, early or mandatory retirement and a lack of career planning were identified. Many participants in the public sector reported shortages of nephrologists as a major problem. Participants in the private sector felt that increasing numbers may allow nephrologists to "*super-specialise*" in a particular field, however others felt that the private sector in major cities was "*saturated*" with nephrologists and that the lack of posts capped the uptake of new graduates in the public sector. There were numerous comments about poor management in the public healthcare sector and views expressed that, in certain provinces, this had resulted in the loss of nephrologists to the private sector.

"*The renal programme in (Province A) is in shambles. Many good people have left the public sector to work in private because the system is not working. The programme is very poorly managed. It is soul destroying.*"

(Public AN, F)

Factors influencing nephrologists' decision in choosing a location of practice included having a bursary from a smaller province and issues around family life ("*schools, children and the family*"), the main consideration for those remaining in the larger cities. The geographic maldistribution resulted in nephrologists in remote areas feeling isolated, overworked and challenged with difficult decision making, which they felt was better conducted in a group. It was noted that paediatric nephrologists in the public sector were more "*active*" than their private sector colleagues, being involved in more challenging aspects of renal care, such as chronic dialysis and transplantation.

A personal reflection of a participant who had emigrated to the United Kingdom and later returned to South Africa provided valuable insights into strong reasons for emigration. Factors that prompted the emigration were: an "*interesting*" career path, more time for research, feeling overworked in South Africa, and concerns regarding family safety. The factors that prompted the return to South Africa included a greater sense of value practicing in South Africa, with a feeling of "*not making a difference*" in the United Kingdom. This feeling of increased individual

value and a personal sense of belonging in South Africa was also expressed by several other participants.

In the private sector, early retirement due to burnout was frequently mentioned. This contrasted with the public sector view, where forced retirement at age 65 deprived nephrologists of the opportunity to work despite them being willing to do so and depriving the nephrology community of many experienced clinicians, researchers and teachers. This was raised by both the younger and senior nephrologists in the public sector group, with younger nephrologists expressing the importance of "*retaining wisdom and teaching*".

"*Even at 65, you've got a good ten years of teaching ability*"

(Public PN, F)

"*Enough is enough.*"

(Private AN, M)

"*Don't think I would have the energy to work beyond that [65 years].*"

(Private AN, M)

A lack of career planning was also identified as a sub-theme in the private sector discussion, with comments of "*no exit strategy*" being in place to ease the transition after completing their formal nephrology training. There was much apprehension about post-training job prospects. A similar lack of direction was reported for the peri-retirement period. One solution mentioned was to encourage mentoring relationships between the younger and the more experienced nephrologists.

"*. . . you see somebody older than you sort of strutting around in the wards, and you say I never want to do that. But for personal planning, that is not something we have. There is no exit plan for us.*"

(Private AN, M)

An increase in the number of nephrologist posts as well as the nephrology training posts in the public sector was recommended to increase the overall numbers, especially with an expected increase in patients accessing RRT after the implementation of the proposed National Health Insurance (NHI) (2), which aims to ensure access to essential health services for all South Africans.

**Theme 3: Shortcomings in nephrology training.**   A few participants felt that "*no real training*" occurred at certain institutions. Low rates of renal transplantation also contributed to poor training at some centres. In centres performing more transplants, a lack of exposure to the mechanics of deceased donor organ allocation was mentioned. Access procedures such as peritoneal dialysis catheter and tunnelled haemodialysis catheter insertions were also identified as being a skills deficit and a lack of training in the technical aspects of haemodialysis was mentioned repeatedly.

"*Similarly, some of the issues around problems with RO [reverse osmosis], management of staff, equipment, technical aspects, some of those things you never learn until you are sitting in a dialysis unit*."

(Private AN, M)

Proficiency at performing renal biopsies was thought to be unnecessary by some private sector nephrologists as the procedure was mainly performed by radiologists in their settings. This was a minority view, however. One participant stated that training in the area would assist private sector nephrologists in identifying and managing complications arising from the procedure. Ultrasound training was emphatically deemed essential by one participant and a "*lovely*" skill by another.

> "*Ultrasound has gone from you kind of maybe had one in the unit ten years ago, now it's the standard of care for every patient. I wheel the ultrasound around with me when I do ward rounds.*"

(Private AN, M)

Training related to non-clinical practice areas, including human resource and asset management, procurement procedures and patient advocacy was also reported as being lacking.

There were divergent views on the value of international fellowships, with most mentioning a benefit to broaden "*perspectives*" and "*views*" and to gain expertise to address local problems ("*lack of transplants*"). Others cited the good quality of local training, making the potential benefit relatively small. Spending time in the well-resourced private sector, as opposed to going abroad, was also suggested to increase "*exposure*". This subtheme of widening training platforms also extended to involving the private sector in more research and training, as this sector was larger in term of numbers of nephrologists as well as patients on dialysis. This was not mentioned in the public sector discussion.

> "*In fact, I think it's going to be essential that the private sector must be involved in research. It's unthinkable that in a setting where 80% of the patients are receiving dialysis for example, that there is no research. It doesn't make sense.*"

(Private AN, M)

A cautionary note on training was made that whilst there was interest in incorporating newer technologies, teaching the "*basics*" of renal medicine should not be undervalued.

**Theme 4: Excessive workload with inadequate remuneration.**   A heavy workload, lack of time for research and inadequate remuneration were aspects identified in both the survey and focus group interviews. Factors contributing to the heavy workload include concerns about financial insecurity and an obligation to service patients. The heavy workload caused some nephrologists to feel "*close to breaking*" and consider early retirement.

> "*It is an insecurity in the beginning because you are starting out and it's not feasible to say no to everything, and then you get onto that treadmill and you can't jump off it. And then you also feel kind of obligated to the patient, and you don't want to turn them away.*"

(Private AN, F)

Participants in both the private and public sector groups reported research being performed outside of working hours. Some private nephrologists mentioned doing research at the cost of losing practice income.

> "*I do enjoy the research side, but the problem is time. . .the only way I have managed to do that is actually to cut my clinical work, and I just have days which I just don't earn.*"

(Private AN, M)

Lower reimbursement for nephrologists compared to other specialities for performing the same procedures was a source of discontent for participants in the private group. However, the overall feeling was that remuneration was "*fair*".

"*The level of remuneration is not consistent with level of responsibility and complexity of care provided.*"

(Private PN, M)

"*Do you know what the difference is between a nephrologist putting in a dialysis catheter and an anaesthetist putting in a dialysis catheter? If you bill medical aid rates, it's like a couple of hundred Rand. The anaesthetists charge R3,000. They charge outside of medical aid rates . . . so then the patients come and complain to you.*"

(Private AN, M)

**Theme 5: A challenging work environment.** Frequent and extensive comments pertained to this theme; however, there were clear differences between the challenges experienced in the different sectors.

*Public sector challenges.* A chronic lack of resources was repeatedly highlighted by nephrologists working in the public sector. This extended from the rationing of access to RRT, through run-down facilities lacking basic amenities, logistic issues around patient transport, to insufficient theatre time for surgical procedures.

"*We work under a very understaffed environment which is affecting the quality of our work. We always have excessive bed occupancy of over 200% and it's impossible to render a good quality service . . . Just poor working conditions and hospital environment that doesn't allow us to treat patients with dignity.*"

(Public AN, F)

A lack of nephrologist posts in the public sector was strongly emphasized by public sector nephrologists, with some reporting working predominantly as general physicians (internists) and not contributing significantly to nephrology services in their hospitals. A concern was raised by the paediatric nephrologists around one-person departments. This provided no opportunity to "*grow a department*" and there was no grooming of successors or succession planning. The lack of posts was not a country-wide problem, with vacant posts in smaller cities being mentioned; however, few participants from the larger cities seemed aware of these opportunities.

"*As a single consultant of the Paediatric Renal Unit since its inception (total 35 years) the service will collapse if I retire without any immediate replacing consultant.*"

(Public PN, F)

A reported lack of support from hospital management and other medical disciplines also contributed to the perception of a difficult working environment. Reasons suggested for the lack of support from hospital authorities included the high costs of RRT as well as an impression that nephrologists are undervalued. Political interference in decisions regarding access to RRT was problematic in few provinces and there was a general lack of understanding by surgical specialities of the needs of patients with ESRD.

"*I often feel vulnerable and at risk being harmed by patients who have been excluded from the RRT and this is made worse by politicians who randomly overrides the panel decisions and want certain patients to be dialysed irrespective of the exclusion criteria.*"

(Public AN, F)

Bureaucratic policies and cumbersome management decision-making processes were repeatedly expressed as an area of concern for both adult and paediatric nephrologists. In the public sector, the commitment of administrative staff to assist nephrologists was found to be lacking, the processes required to facilitate patient care was unnecessarily complex and time consuming, and legislation hindered older medical practitioners (over the age of 65) from working in the public sector.

"*I can spend three hours on the telephone trying to get a patient to Hospital B for a transplant. I mean, it's ridiculous how they won't book a flight. You have to get permission from the CFO of the province to book a flight.*"

(Public AN, M)

The rationing of dialysis in the public sector created difficult choices for nephrologists regarding which patients get access to therapy.

"*I feel sometimes, because I'm the only nephrologist in the province... everyone looks to me to make the decision, putting someone on dialysis or not, and that, going to bed with that, is troublesome*"

(Public AN, F)

*Private sector challenges.* Concerns from the private sector were centred on the unhelpful nature of the health insurance companies. "*Interference*" from these companies in clinical care was noted and sometimes culminated in harm to patients. Difficulties in getting approvals to fund RRT had "*escalated in the last couple of years*" with many "*unreasonable requests*" for additional information or motivational letters. These administrative tasks consumed a large amount of time and was a major source of frustration. Among the other concerns of private sector nephrologists were a lack of clarity regarding the implementation of the NHI and doubts about its feasibility, dealing with the high expectations of families, and frequent medico-legal issues.

"*Medical aids have adopted the State mechanism of adding bureaucratic obstacles to getting everything done... You must fill in forms for everything now. In fact, in (Insurance Company A), you must fill in the first form to get the second form!*"

(Private AN, M)

"*NHI... will really finish medicine ... Every government department so far has failed in everything they tried to do, and that's what's going to happen with medicine. I can't see how they could make the private sector fund all the deficiencies in the government sector.*"

(Private AN, M)

## Discussion

This study on human resources for nephrology in South Africa is the first in-depth report on the topic. The density of nephrologists in South Africa is low (2.5 pmp), well below the median

for high-middle-income countries (10.8 pmp) [17]. Most nephrologists are concentrated in the private sector which serves 16% of the population. This maldistribution is compounded by regional inequalities, with no nephrologists at all in two of the nine South African provinces. While most nephrologists are happy with their remuneration and working conditions, a substantial proportion are not. Early retirement and emigration are being seriously contemplated by 19% and 11% of the surveyed nephrologists, respectively, and this may exacerbate the deficit in nephrologist numbers.

Prominent positive factors identified were the complex and challenging nature of the discipline, the diversity of clinical practice, the close relationships with patients and the ability to contribute to a drastic improvement in their quality of life.

The challenges identified were more numerous and included an overall shortage of nephrologists and an unequal geographic and sectorial distribution, some shortcomings in training and a heavy workload. In the public sector, nephrologists struggle with difficult decisions due to the rationing of RRT, a severe lack of resources (including posts), and insufficient support from hospital management. In the private sector, nephrologists struggle with the administrative load imposed by medical insurance companies and are concerned about the potential impact of NHI.

The number and density of nephrologists in South Africa is higher than that reported from the previous estimate of 1.1 pmp in 2004 [19], and the more recent estimate of 1.9 pmp reported in 2018 [18]. Whilst the difference from the former figure may be due to growth over time, the difference from the more recent study is likely due to the robust methodology we employed to identify nephrologists, with triangulation of data using several sources.

Even though the numbers have increased, they remain inadequate when compared to other world regions as well as countries of a similar GDP (Figs 4 and 5). At first glance, the figure of 1 nephrologist per 74 patients on RRT compares favourably with the ratio of 1-to-75 recommended by the British Renal Society [25]. However, this ratio is completely skewed by the large numbers of patients with ESRD who are not afforded dialysis and transplantation, but who still require nephrology care. If substantially increased access to RRT is achieved in South Africa, the shortage of nephrologists will be more clearly exposed.

Whilst the geographic maldistribution of nephrologists was mainly due to individual choices on location mainly based on the demands of family life, it remains an issue that requires addressing. Nephrologists in remote areas reported numerous difficulties due to the low number of nephrologists and the high workload. Transportation difficulties and a lack of resources made patient care in these areas more complex. Some nephrologists who were working in under-resourced provinces did so due to contractual obligations towards those who had sponsored their training. This suggests that such policies may increase the numbers of nephrologists settling in under-resourced provinces and should be more widely adopted. Nephrologists in these areas should act as mentors for aspiring nephrologists, as studies have shown that positive role models can attract fellows to pursue a career in nephrology and that exposure to remote areas resulted in more trainees choosing to practice in those areas after completion of training [27, 28]. Other solutions to increase nephrologist numbers, especially in the public sector, would require investment to increase the numbers of specialist nephrologist posts as well as training posts.

Whilst the reasons for considering emigration mainly reflect wider problems in South African society, it is notable to reflect on the sense of value and belonging felt by most nephrologists in South Africa when they were asked about emigration.

The trends also show a positive increase in the number of women and a younger cadre trained as nephrologists. This is likely the result of concerted efforts by medical schools to increase their intake of female medical students. The feminisation of the medical profession is

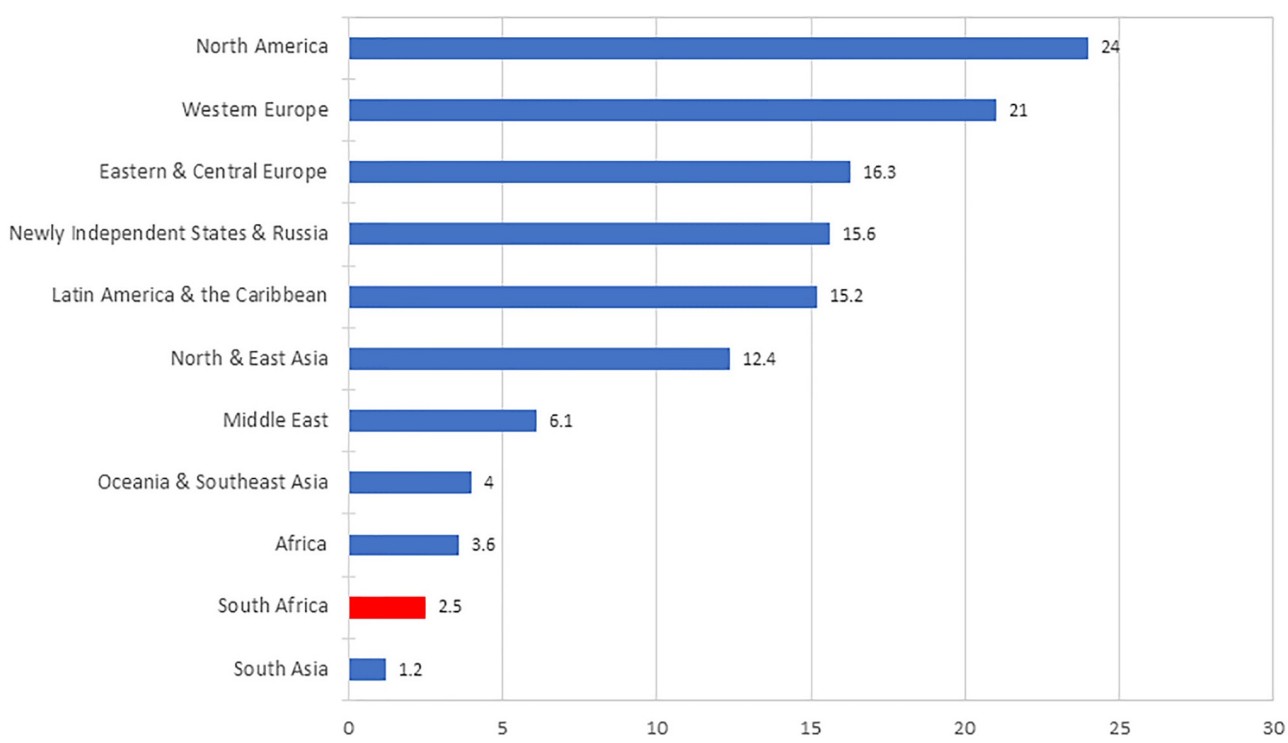

**Fig 4. Density of nephrologists in various world regions and South Africa.** From Osman et al. [18]. Data for South Africa is from the present study. Abbreviations: pmp, per million population.

a topic of much discussion and has implications for the size of the workforce required to accommodate the transition from full-time to part-time work during child-bearing years.

In the public sector, labour legislation and the lack of part-time posts make working beyond the age of 65 years an unlikely prospect. A solution which includes amendments to the legislation and offers flexible working hours is required. In the private sector, early retirement is also a concern for loss of an active part of the workforce. Solutions that may mitigate this trend include increasing the number of nephrologists being trained, working arrangements which permit more time off, and minimising the administrative burden from health insurance companies.

Our study confirmed the disparity in human resources between the private and public sectors in South Africa. During the focus group interviews, a willingness was expressed by two private sector nephrologists to assist with roles traditionally performed by the public sector, including teaching and research. However, in the survey data, we found close to no time spent on research or teaching by any private sector nephrologist. Strong concerns were raised around the implementation of the NHI bill by the private sector nephrologists. Areas of uncertainty include the extent to which private financing mechanisms continue to pay for RRT, and the extent to which the NHI fund will purchase services from private providers. Nephrologists felt that the National Department of Health should actively engage the South African Renal Society in drafting the plans for the NHI's renal services. Failure to do so may strengthen many nephrologists' resolution to emigrate.

An interesting finding was the greater number of paediatric nephrologists in the public sector, which was described as more active in ESRD management than the paediatric private

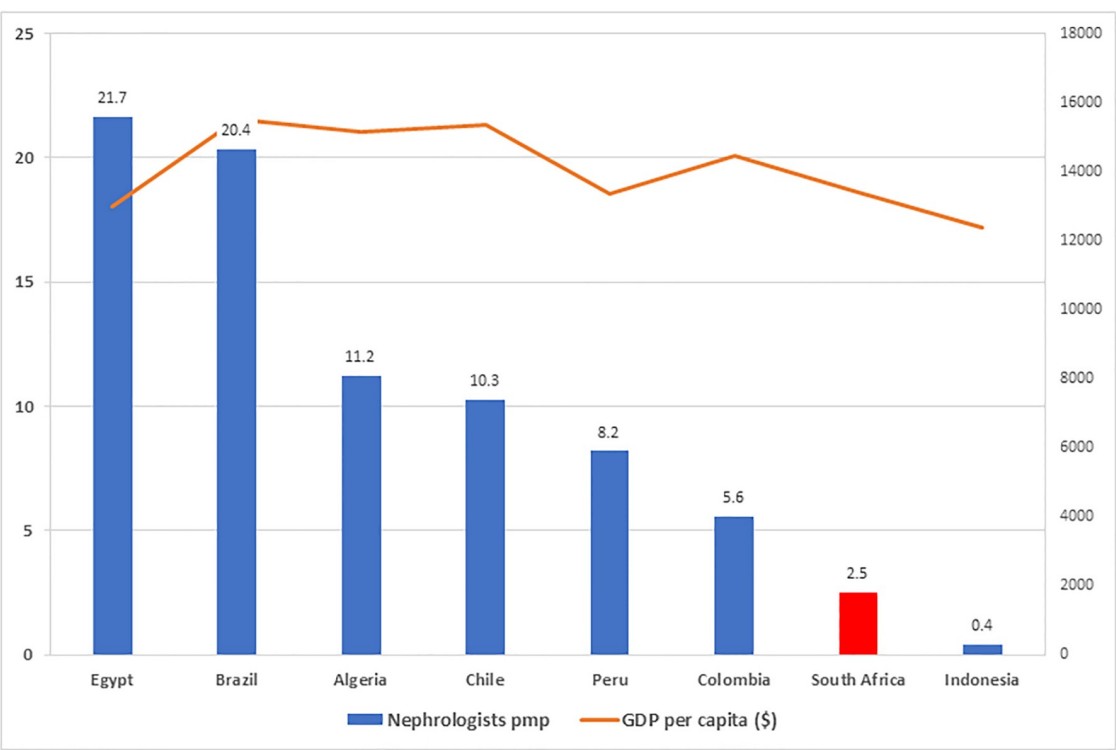

**Fig 5. Density of nephrologists amongst countries with similar gross domestic product per capita to South Africa.** From Osman et al. [18] and World Economic Outlook Database [26]. Data for South Africa is from the present study. Abbreviations: GDP, gross domestic product per capita in 2017 by the purchasing power parity method (in 2011 international dollars); PMP, per million population.

sector. The absolute number of paediatric nephrologists in South Africa is small, and concentration of these highly trained individuals around public sector teaching hospitals is a likely explanation for this. Some teaching centres have only a single paediatric nephrologist, running an important service which may collapse if long-term planning and sustainable measures are not implemented.

While most nephrologists report being proficient and having adequate training for the traditional skills required of nephrologists, training on the use of ultrasound to evaluate renal anatomy and vascular access are aspects where there is room for improvement. A substantial number of nephrologists report self-initiated training in this domain. Further training in the technical aspects of haemodialysis, peritoneal dialysis catheter insertion as well as management skills to run a dialysis unit or a clinical department are needed. Even though many nephrologists in the private sector do not perform renal biopsies in their practice, it is still seen as an essential competency for nephrologists in both health sectors. These data also highlight the array of skills of South African nephrologists, with higher self-reported competency rates at procedures compared to other countries [29, 30].

Our findings on nephrologists' perceptions of their chosen discipline are largely in keeping with those described by others, with certain aspects unique to South Africa [14, 27, 29]. These include the challenges in making decisions about access to RRT, a propensity to consider emigration due to wider social problems, frustrations borne from bureaucratic burdens, and apprehension about the effects of NHI on the private sector.

Some of the strengths of our study include the use of triangulation to enhance trustworthiness and the credibility of our data analysis. Multiple data sources were used to identify with

greater precision all active nephrologists in South Africa, and the use of both quantitative and qualitative methods allowed us to verify data and to obtain an in-depth understanding of our nephrologist workforce. Limitations include our formal definition of a nephrologist, which excludes specialist physicians who are not registered as nephrologists but have an interest in nephrology and look after patients with CKD/ESRD. Another limitation was our inability to recruit a paediatric nephrologist for the private sector focus group interview. Of the 16 paediatric nephrologists who responded to the REDCap survey, 4 were in private practice, mitigating this limitation to some extent.

## Conclusions

There are insufficient numbers of nephrologists in South Africa, and a markedly uneven distribution amongst the different provinces and healthcare sectors. South African public sector nephrologists are faced with the challenges of considerable resource limitations, a high workload, challenging policies and unsatisfactory remuneration. In the private sector, nephrologists face a high workload, large administrative burdens and uncertainty about the potential impact of NHI. A substantial proportion of the workforce is contemplating emigration or early retirement.

This data should assist in workforce planning and guide policy development to alleviate the challenges faced by nephrologists in South Africa. In the public sector, more resources need to be allocated to renal care and the disparities between provinces should be actively addressed. Training institutions need to implement some modifications to their nephrology programmes and medical insurance companies need to reduce the administrative burden placed on nephrologists in the private sector.

## Supporting information

**S1 Appendix. Survey questionnaire.**
(PDF)

**S2 Appendix. Prompts used during the focus group discussions.**
(DOCX)

**S1 Table. Demographic details of survey respondents (n = 81) working in the public and private healthcare sectors.** Abbreviations: IQR, interquartile range.
(DOCX)

**S1 Fig. a**. Percentage of adult nephrologists (n = 66) reporting competency at various procedures, attained either during formal nephrology training or via self-initiated training. Abbreviations: HD, haemodialysis; PD, peritoneal dialysis. **b**. Percentage of paediatric nephrologists (n = 15) reporting competency at various procedures, attained either during formal nephrology training or via self-initiated training. Abbreviations: HD, haemodialysis; PD, peritoneal dialysis.
(ZIP)

## Acknowledgments

We thank our colleagues in the nephrology community for generously giving of their time to participate in the study.

## Author Contributions

**Conceptualization:** Adriano Pellizzon, Usuf M. E. Chikte, Mogamat Razeen Davids.

**Data curation:** Muhammed Hassen, Adriano Pellizzon, Usuf M. E. Chikte, Mogamat Razeen Davids.

**Formal analysis:** Muhammed Hassen, Elize Archer, Mogamat Razeen Davids.

**Investigation:** Muhammed Hassen, Adriano Pellizzon, Mogamat Razeen Davids.

**Methodology:** Elize Archer, Mogamat Razeen Davids.

**Supervision:** Usuf M. E. Chikte, Mogamat Razeen Davids.

**Writing – original draft:** Muhammed Hassen, Mogamat Razeen Davids.

**Writing – review & editing:** Muhammed Hassen, Elize Archer, Adriano Pellizzon, Usuf M. E. Chikte, Mogamat Razeen Davids.

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
