## [Decision Letter · Decision Letter 0]

19 Nov 2019

PONE-D-19-29781

Human resources for nephrology in South Africa: a mixed-methods study

PLOS ONE

Dear Prof Davids,

Thank you for submitting your manuscript to PLOS ONE. After careful consideration, we feel that it has merit but does not fully meet PLOS ONE’s publication criteria as it currently stands. Therefore, we invite you to submit a revised version of the manuscript that addresses the points raised during the review process.

Both reviewers were generally positive regarding this submission. There are a couple of points which require revision to ensure that the submission fulfils the criteria required for a qualitative study (http://journals.plos.org/plosone/s/submission-guidelines#loc-qualitative-research). The authors do not clearly lay out their objectives in performing the study. In addition, the methodology is brief and the rationale for the survey design is not completely clear. Finally, there is a suggestion of bias which is not discussed in the discussion. This includes the fact that all of the paediatric nephrologists who were interviewed were from the public sector for example and the fact that purposive and convenience sampling was required.

We would appreciate receiving your revised manuscript by Jan 03 2020 11:59PM. To enhance the reproducibility of your results, we recommend that if applicable you deposit your laboratory protocols in protocols.io, where a protocol can be assigned its own identifier (DOI) such that it can be cited independently in the future. For instructions see: http://journals.plos.org/plosone/s/submission-guidelines#loc-laboratory-protocols

We look forward to receiving your revised manuscript.

Kind regards,

Elizabeth S. Mayne, M.D.

Academic Editor

PLOS ONE

Journal Requirements:

1. Please provide additional details regarding participant consent. In the ethics statement in the Methods and online submission information, please ensure that you have specified (1) whether consent was suitably informed and (2) what type you obtained (for instance, written or verbal).

Additional Editor Comments (if provided):

Both reviewers were generally positive regarding this submission. There are a couple of points which require revision to ensure that the submission fulfils the criteria required for a qualitative study (http://journals.plos.org/plosone/s/submission-guidelines#loc-qualitative-research). The authors do not clearly lay out their objectives in performing the study. In addition, the methodology is brief and the rationale for the survey design is not completely clear. Finally, there is a suggestion of bias which is not discussed in the discussion. This includes the fact that all of the paediatric nephrologists who were interviewed were from the public sector for example and the fact that purposive and convenience sampling was required.

Reviewers' comments:

Reviewer's Responses to Questions

**Comments to the Author**

1. Is the manuscript technically sound, and do the data support the conclusions?

Reviewer #1: Yes

Reviewer #2: Yes

2. Has the statistical analysis been performed appropriately and rigorously? 

Reviewer #1: Yes

Reviewer #2: Yes

3. Have the authors made all data underlying the findings in their manuscript fully available?

Reviewer #1: Yes

Reviewer #2: Yes

4. Is the manuscript presented in an intelligible fashion and written in standard English?

Reviewer #1: Yes

Reviewer #2: Yes

5. Review Comments to the Author

Reviewer #1: Proposed Changes:

Page 2: Methods - Define what is meant by distribution, e.g. geographic, provincial, etc.

Page 2: Results - For the median age, add the interquartile range that is reported in results.

Page 3: Second last paragraph - Delete in South Africa (already stated in earlier sentence

Page 3: Conclusion - What standard was used to determine that we insufficient nephrologists. Is this based on a global standard?

Page 6: Introduction - Add correct punctuation for therefore

Page 8: Correct the verb

Original text

The interviews were conducted in English, used open-ended questions and prompts (S2 Appendix) to initiate the iscussion,

and avoided the use of leading questions

correct to using open-ended

Page 9: Results

Remove paragraph

“These findings point to increasing training of female nephrologists and

those of African or Indian/Asian ancestry in recent years.”

Page 9: Results – Quantitative data

Original sentence

"Most nephrologists were South African citizens (92%, n=130), while six were citizens of other African countries, four were

citizens of European countries and two were citizens of countries in Asia."

Add the percentages for the nephrologists from African and European countries

Page 9: Table 1 Consider using Black African rather than Black. Also consider aligning data to the right,

Page 7: Methods -Please describe the defined race groups as per the South African Population Registration Act (Act No.30 of 1950). For example, the use of mixed race should be defined as referring to coloured n this act.

Page 11: Table 2 -Also consider aligning data to the right,

Reviewer #2: The authors have made a good attempt to achieve a good response rate, which makes the data more valid. The article is well laid out. The subject matter is important in both a local and international context. No grammatical or typographical errors were noted.

6. PLOS authors have the option to publish the peer review history of their article (what does this mean?). If published, this will include your full peer review and any attached files.

Reviewer #1: No

Reviewer #2: Yes: Graham Paget

---

## [Author Response · Author response to Decision Letter 0]

25 Dec 2019

Editor comments:

1. There are a couple of points which require revision to ensure that the submission fulfils the criteria required for a qualitative study. The authors do not clearly lay out their objectives in performing the study. In addition, the methodology is brief and the rationale for the survey design is not completely clear. 

Response: Thank you for referring us to the COREQ and SRQR checklists.

The last paragraph of the Introduction has been expanded to provide more clarity on the objectives/rationale of the study.

The Methods section has been expanded:

We have added a paragraph at the start of the Methods to provide detail on the research team, their backgrounds and assumptions. 

We specify who did the recruitment and managed the survey.

We have elaborated on the focus group interviews.

We have elaborated on the protection of the privacy of participants at the end of the Methods section.

We have expanded on the strengths and limitations paragraph at the end of the Discussion.

2. There is a suggestion of bias which is not discussed in the discussion. This includes the fact that all the paediatric nephrologists who were interviewed were from the public sector and the fact that purposive and convenience sampling was required.

Response: This is now included in the strengths and limitations paragraph at the end of the Discussion.

3. Please provide additional details regarding participant consent. In the ethics statement in the Methods and online submission information, please ensure that you have specified (1) whether consent was suitably informed and (2) what type you obtained (for instance, written or verbal).

Response: These details are now provided in the ethics statement at the end of the Methods section.

Reviewer 1:

4. Page 2, Methods: Define what is meant by distribution, e.g. geographic, provincial, etc.

Response: This has been clarified to place the emphasis is on province.

5. Page 2, Results: For the median age, add the interquartile range that is reported in results.

Response: Done.

6. Page 3, second-last paragraph: Delete in South Africa (already stated in earlier sentence)

Response: We have not changed the figure title as this needs to stand on its own.

7. Page 3, Conclusion: What standard was used to determine that we have insufficient nephrologists. Is this based on a global standard?

Response: Yes, we used international benchmarking. The Global Kidney Health Atlas reports the median nephrologist density for high-middle-income countries as 10.8 pmp. We have now established that the density in South Africa is 2.5 pmp, well below this level. We make this point at the beginning of the Discussion. The only other benchmark we are aware of is the British Renal Association recommendation of 1 nephrologist for every 75 patients on renal replacement therapy (RRT). It would not be appropriate to use this ratio for countries (like South Africa) where the number of patients on RRT is artificially low because of limited resources and poor access to RRT; in other words, where the number of patients on RRT does not reflect the overall burden of renal disease in the country.

8. Page 6, Introduction: Add correct punctuation for therefore

Response: Done.

9. Page 8: Correct the verb

Response: Done.

10. Original text: The interviews were conducted in English, used open-ended questions and prompts (S2 Appendix) to initiate the discussion, and avoided the use of leading questions. Correct to “using open-ended”.

Response: Done.

11. Page 9, Results: Remove paragraph - “These findings point to increasing training of female nephrologists and those of African or Indian/Asian ancestry in recent years.”

Response: Done.

12. Page 9, Results – Quantitative data: Original sentence "Most nephrologists were South African citizens (92%, n=130), while six were citizens of other African countries, four were citizens of European countries and two were citizens of countries in Asia." Add the percentages for the nephrologists from African and European countries.

Response: Done.

13. Page 9, Table 1: Consider using Black African rather than Black. Also consider aligning data to the right.

Response: Done.

14. Page 7, Methods: Please describe the defined race groups as per the South African Population Registration Act (Act No.30 of 1950). For example, the use of mixed race should be defined as referring to coloured in this act.

Response: Done. The apartheid government created four official racial categories using the Population Registration Act: Black, Coloured (mixed ancestry), White and Asian/Indian. Today, Statistics South Africa, which is the national statistical service of South Africa, asks citizens to self-classify into one of five categories: Black African, Coloured, White, Indian/Asian and Other/Unspecified.

15. Page 11, Table 2: Also consider aligning data to the right

Response: Done.

Additional changes:

We have updated the manuscript with the recently published 2017 data from the SA Renal Registry (reference 6).

Several minor typographical and grammatical errors gave been fixed.

We mention that no external funding was received for the study.

---

## [Decision Letter · Decision Letter 1]

27 Jan 2020

Human resources for nephrology in South Africa: a mixed-methods study

PONE-D-19-29781R1

Dear Dr. Davids,

We are pleased to inform you that your manuscript has been judged scientifically suitable for publication and will be formally accepted for publication once it complies with all outstanding technical requirements.

With kind regards,

Elizabeth S. Mayne, M.D.

Academic Editor

PLOS ONE

Additional Editor Comments (optional):

Reviewers' comments:

Reviewer's Responses to Questions

**Comments to the Author**

1. If the authors have adequately addressed your comments raised in a previous round of review and you feel that this manuscript is now acceptable for publication, you may indicate that here to bypass the “Comments to the Author” section, enter your conflict of interest statement in the “Confidential to Editor” section, and submit your "Accept" recommendation.

Reviewer #1: All comments have been addressed

2. Is the manuscript technically sound, and do the data support the conclusions?

Reviewer #1: Yes

3. Has the statistical analysis been performed appropriately and rigorously? 

Reviewer #1: Yes

4. Have the authors made all data underlying the findings in their manuscript fully available?

Reviewer #1: Yes

5. Is the manuscript presented in an intelligible fashion and written in standard English?

Reviewer #1: Yes

6. Review Comments to the Author

Reviewer #1: Page 8

Reword (nine nephrologists) and (two nephrologists) as (n=9) and (n=2)

Page 9 Reword “Analysis of the quantitative data was performed in Microsoft Excel”

To

"Analysis of the quantitative data was performed using Microsoft Excel (Redmond, WA, USA)"

Page 11 Use of Black and African used in results. Rather refer to Black African for consistency.

Page 14 Reword four of the five

To 4/5

Page 24 Reword”

“,with no nephrologist at all in two of the nine”

To

“,with no nephrologist in two”

Page 3 Remove extra space

“an overall shortage of”

7. PLOS authors have the option to publish the peer review history of their article (what does this mean?). If published, this will include your full peer review and any attached files.

Reviewer #1: No

---

## [Editor Report · Acceptance letter]

29 Jan 2020

PONE-D-19-29781R1 

Human resources for nephrology in South Africa: a mixed-methods study 

Dear Dr. Davids:

I am pleased to inform you that your manuscript has been deemed suitable for publication in PLOS ONE. Congratulations! Your manuscript is now with our production department. 

With kind regards,

on behalf of

Dr. Elizabeth S. Mayne 

Academic Editor

PLOS ONE